# Quantitative Real-Time PCR Based on SYBR Green Technology for the Identification of *Philaenus italosignus* Drosopoulos & Remane (Hemiptera Aphrophoridae)

**DOI:** 10.3390/plants11233314

**Published:** 2022-11-30

**Authors:** Domenico Rizzo, Matteo Bracalini, Sara Campigli, Anita Nencioni, Francesco Porcelli, Guido Marchi, Daniele Da Lio, Linda Bartolini, Elisabetta Rossi, Patrizia Sacchetti, Tiziana Panzavolta

**Affiliations:** 1Laboratory of Phytopathological Diagnostics and Molecular Biology, Tuscany Regional Plant Health Service, Via Ciliegiole 99, 51100 Pistoia, Italy; 2Department of Agricultural, Food, Environmental and Forestry Science and Technology (DAGRI), University of Florence, Piazzale delle Cascine 28, 50144 Florence, Italy; 3Department of Soil Sciences, of Plants and Food (Di.S.S.P.A), University of Bari Aldo Moro, 70121 Bari, Italy; 4Department of Agriculture, Food and Environment (DAFE), University of Pisa, Via del Borghetto, 80, 56124 Pisa, Italy

**Keywords:** *Xylella fastidiosa* vector, alpha taxonomic identification, molecular diagnostic tool, spittlebug

## Abstract

The use of molecular tools to identify insect pests is a critical issue, especially when rapid and reliable tests are required. We proposed a protocol based on qPCR with SYBR Green technology to identify *Philaenus italosignus* (Hemiptera, Aphrophoridae). The species is one of the three spittlebugs able to transmit *Xylella fastidiosa* subsp. *pauca* ST53 in Italy, together with *Philaenus spumarius* and *Neophilaenus campestris*. Although less common than the other two species, its identification is key to verifying which role it can play when locally abundant. The proposed assay shows analytical specificity being inclusive with different populations of the target species and exclusive with non-target taxa, either taxonomically related or not. Moreover, it shows analytical sensibility, repeatability, and reproducibility, resulting in an excellent candidate for an official diagnostic method. The molecular test can discriminate *P. italosignus* from all non-target species, including the congeneric *P. spumarius*.

## 1. Introduction

The spittlebug *Philaenus italosignus* Drosopoulos & Remane (Hemiptera, Aphrophoridae) is a Mediterranean species, endemic to Italy, recently demonstrated to be a competent vector of the pathogenic bacterium *Xylella fastidiosa* subsp. *pauca* ST53 together with two other aphrophorid species: *Philaenus spumarius* (L.) and *Neophilaenus campestris* (Fallen) [1]. Up to 2019, the known distribution range of *P. italosignus* was restricted to a few Italian areas with unconnected records along central-southern Italy, including Sicily [2]. Recently, *P. italosignus* has been reported on its host plant, *Asphodelus ramosus* L. [3], in a coastal area relatively close to Monte Argentario, southern Tuscany, where *Xylella fastidiosa* subsp. *multiplex* ST87 was also recently reported [4,5]. Moreover, adults of *P. italosignus* were also observed on olive trees at the same site [3]. Usually, *P. italosignus* is less abundant than *P. spumarius* in olive groves, so it is considered to play a secondary role, compared to the meadow spittlebug, in transmitting *X. fastidiosa*. However, it could become the primary vector of this bacterium in other agricultural and natural ecosystems [1]. Thus, to plan effective control strategies and limit the spread of X. fastidiosa, all vectors, including less abundant species, have to be monitored.

Currently, the identification of European vectors of *X. fastidiosa* is mainly based on morphological traits. However, new molecular protocols have been recently proposed in an effort to identify both the vectors and the pathogen. Main morphological features of adult specimens are used as diagnostic characteristics (e.g., size, shape, frons features, processes on the hind tibia), though species diagnosis requires the accurate observation of male genitalia performed by expert personnel [6]. Finally, color polymorphism may be useful to discriminate *P. spumarius*, at least in the case of the exclusive morph patterns. On the other hand, in *P. italosignus,* this is never a reliable diagnostic feature, as its morphological patterns are all shared with *P. spumarius* [7]. Undoubtedly, the availability of diagnostic molecular methods would help to make the surveillance of these species easier and faster, especially in the case of expected disease outbreaks. A protocol for DNA barcoding in the *Philaenus* genus was defined to study the evolutionary relationships among species and near families [8]. Also, two protocols using conventional and quantitative PCR to identify *P. spumarius* were recently proposed [6]. However, no specific assay for the identification of *P. italosignus* has been established yet. In this paper, a molecular diagnostic assay based on SYBR Green technology has been set up to allow the species identification of *P. italosignus*. Moreover, whenever the presence of *X. fastidiosa* has to be verified in the field, the proposed assay could be of paramount importance for the simultaneous identification of both the vector and the bacteria in field mass screenings.

## 2. Results

### 2.1. DNA Extraction

The results of the DNA extraction are shown in Table 1, where the average concentrations (ng/µL) of DNA extracted with the respective standard deviations (SD), the absorbance ratios (A260/280), and the Cq values (quantification cycles) obtained using qPCR for testing the 18S ribosomal gene amplifiability are shown [9].

### 2.2. Development and Optimization of P. italosignus-Specific qPCR

Database searches and sequence alignments indicated both heterozygosity and the existence of insertion/deletions in the intron region of the EF-1α gene of *P. spumarius* PSPU_08 (GCA_018207615.1). PCR amplification performed with EF-1αPCR primers set Spit-Uni1/Spit-Uni2 yielded a fragment of the expected size (approx. 700 bp) from both *P. italosignus* and *P. spumarius.* Still, it failed to amplify the DNA of *N. campestris*. Although direct sequencing chromatograms had to be discarded because of superimposed mixed traces, the sequencing of the cloned fragments was successful and indicated that the target gene had been amplified (data not shown). Moreover, the alignment of at least five inserts from each amplicon revealed the existence in all four genomes of at least two EF-1α heterozygous alleles due to SNPs and insertions/deletions (indels) between intron sequences. One of these indels, being conserved within and between genomes of the same insect species, was exploited to design the *P. italosignus* EF-1α alleles-specific primer Pital_465R (Figure 1).

qPCR optimal reaction mix in a final volume of 20 µL was obtained with 10 µL of 2× QuantiNova Probe PCR Master Mix (Qiagen, Hilden, Germany) with 0.4 µM of primer concentration. The optimal annealing temperature for the qPCR reaction was 58 °C. The final PCR reaction in two steps had a melting peak of around 73 ± 0.5 °C (Figure 2). Samples were considered positive when the resulting qPCR curves showed a clear inflection point (in addition to increasing kinetics), and Cq values were <35 with a defined melting peak, as previously reported. No amplification was obtained from *P. spumarius* and *N. campestris* samples.

### 2.3. Performance Characteristics

All the assays performed resulted inclusive for *P. italosignus* and exclusive for the non-target organisms tested. The target specimens were correctly identified using the specific tests, and no false positives (with reference to the cut-off value for the qPCR assay) were obtained with non-target organisms. Thus, both inclusivity and exclusivity resulted in 100% analytical specificity of the test. qPCR runs yielded the same qualitative results and were not affected by variations in testing conditions. The analytical sensitivity (LoD) was evaluated on adults of *P. italosignus* and was equal to 0.016 ng/µL (Table 2). The *R^2^* correlation values were equal to 0.99 for the assayed adults of *P. italosignus* (Figure 3). The repeatability and reproducibility are shown in Table 3. The repeatability values ranged between 26.57 ± 0.01 and 26.94 ± 0.24, while for the reproducibility, they were between 26.45 ± 0.24 and 26.98 ± 0.28.

### 2.4. Field Protocol Validation

A total of 43 *Philaenus* adults were collected in the Orbetello area, 17 from the December sampling and 26 from the June sampling (Table 4). In December, morphological observations led to the identification of 15 *P. spumarius* adults (five males and 10 females, 10 of which were collected directly from the lily (five males and five females). Only two specimens of *P. italosignus* were found during the December sampling (both females). In contrast, all the specimens from the June sampling, collected with sweeping nets on both tree crowns and herbaceous vegetation, were identified as *P. italosignus* (11 males and 14 females), except one male of *P. spumarius*. Our diagnostic assay confirmed the identification of every *P. italosignus* specimen without cross-reaction in *P. spumarius*.

## 3. Discussion

Identification at the specific level of spittlebug vectors of *X. fastidiosa* is key to the development of new strategies for monitoring and managing this serious phytosanitary emergency [6,10,11,12]. Despite previous attempts to develop a molecular diagnostic tool for the species of *Philaenus* genus [8,13], currently available molecular protocols are not specific to *P. italosignus* [6]. In this paper, we present a novel SYBR Green qPCR assay specific to *P. italosignus* that seems very advantageous in terms of reliability and reproducibility when coupled with an efficient DNA extraction protocol [14]. Also, real-time assays don’t require post-PCR electrophoresis, reducing the time of analyses as well as residual waste [15]. In addition, our qPCR SYBR Green assay offers cost advantages also as it is almost 50% cheaper than other diagnostic tools like qPCR Probre or Real-time Lamp. In fact, the cost for reagents and other consumables needed for one sample reaction would amount to about 5 euros, not taking into account DNA extraction. These features are really advantageous in routine monitoring when a large amount of samples needs to be tested every day, implying higher safety standards for operators. At the same time, in the case of a low number of target spittlebugs, this molecular approach may be advantageous. In fact, low populations of P. *italosignus* may result in an even lower number of sampled male adults. Thus, a molecular test may exploit all specimens, including females, that cannot be as easily identified morphologically.

Our qPCR assay was demonstrated to be specific for the target, allowing its unambiguous identification from non-target species, either not taxonomically related to *P. italosignus* or very close to it, like *P. spumarius* and *N. campestris*. This is in agreement with other studies where specific SYBR Green qPCR assays were developed for the identification of insect pests, also considering congeneric non-target species [15,16]. High specificity granted by this technology may be useful when dealing with lesser-studied insects in all fields of application, as in the case of non-pest species (e.g., *Philaenus* spp.), which may become vectors of important plant pathogens. Then, the degree of uncertainty typically leading to the erroneous identification of insect species may be overcome by highly specific SYBR Green qPCR assays [15].

Our assay exploits the existing indels between *P. italosignus* and *P. spumarius* in a 119 bp intron region of the EF1-α gene. Since the presence of a high intrataxon and, possibly intragenomic, variability in this non-coding region has been found in *P. spumarius* [17], to develop novel qPCR-specific primers, we verified if this was also true for *P. italosignus*. Indeed, we confirmed the existence of at least two EF-1α alleles in *P. italosignus* and *P. spumarius* Italian specimens and exploited this heterozygosity to design Pital_348F/465R primers. Given the limited number of *P. spumarius* and *P. italosignus* specimens whose EF-1α gene alleles were sequenced during the heterozygosity exploration, we confirmed the absence of cross-reaction in *P. spumarius* by taking into account the recently found mixed population of both spittlebugs from Orbetello, that was sampled from different plant species and in different seasons (December and June). By doing so, we could always test composite samples in which either one or the other spittlebug was dominant according to morphological analysis. Nonetheless, analyzing a total of 43 specimens, interference with the assay′s specificity was never observed. Follow-up studies may be carried out on a larger number of samples of both adult and juvenile specimens to further validate the protocol.

Data presented here indicate that this protocol is suitable for field massive sampling programs aimed at defining the role of different spittlebug vectors in spreading *X. fastidiosa* in different environments, plant hosts, and seasons. Indeed, to be able to precisely assess the presence of *P. italosignus* in a given area, as we found to be the case also in Tuscany, could be of significant importance since its earlier activity, compared to *P. spumarius*, implies a broader timeframe for potential bacterium transmission [3].

## 4. Material and Methods

### 4.1. Insect Samples for the Setup of the Molecular Diagnostic Protocol

Adults of *P. italosignus* were collected in the Maremma Regional Park (42.670928N; 11.085984E), where a population of the spittlebug was first observed in Tuscany in 2019. The specimens were collected with sweeping nets or mouth aspirators and then examined in the laboratory to confirm their morphological identification. *Philaenus spumarius* and other non-target Auchenorrhyncha were collected throughout Tuscany as part of the area surveillance activities from 2017 to 2020 by the University of Florence and the Phytosanitary Service of the Tuscany Region [18]. All specimens were killed in 96% ethanol and examined in the laboratory to confirm their identification based on morphological features.

Male *Philaenus* adults were identified by examining the genitalia [2,6]. Also, comparing our data with those available in the literature for *P. italosignus* and *P. spumarius,* total body length was used as a diagnostic feature to identify *Philaenus* females [2,3,6]. Adults were killed and preserved in 96% ethanol until processed. Non-target specimens were either adults or preimaginal stages of several species collected in the field or mass-reared, as well as DNA sequences available in the molecular insect collection at the phytopathological laboratory of the Regional Phytosanitary Service (RPS) of the Tuscany Region in Pistoia (Tuscany, Italy).

Target and non-target species are listed in Table 5: the species of Aphrophoridae assayed included the target species *P. italosignus* and the other two vectors of *X. fastidiosa, P. spumarius* and *N. campestris*). To test the exclusivity of the assay, some species taxonomically unrelated to *P. italosignus* were also chosen.

### 4.2. Heterozygosity of the Elongation Factor-1α Gene in Philaenus italosignus and P. spumarius

Elongation Factor-1α (EF-1α) was chosen as the candidate gene for discriminating *P. italosignus* from *P. spumarius* through qPCR. The homologous sequences of *P. italosignus* (JF309079) and *P. spumarius* PSPU_08 (GCA_018207615.1) were retrieved from the GenBank database (https://www.ncbi.nlm.nih.gov/genbank/; accessed on 1 November 2021) and aligned using Muscle as implemented in the Mega11 software package [19]. To verify the existence of heterozygosity in the two species, primers Spit-Uni1 (5′-ACTAGTAAGCCCCAGTGAAATCA-3′) and Spit-Uni2 (5′-ACAAATGCGGTGGTATCGAY-3′) were designed to amplify a fragment of approximately 700 bp. The reagent mix composition was as described herein: 1 X GreenTaq PCR Buffer (Thermo Scientific, Waltham, MA, USA), 0.2 mM of each dNTP, 1.25 U of DreamTaq (Thermo Scientific, Waltham, MA, USA), 0.2 µM of each primer (Eurofins Scientific, Italy), and c. 20 ng of DNA extracted as indicated below from two *P. italosignus* (Pi1 and Pi2) and two *P. spumarius* (Ps3 and Ps4) adults, sampled in different areas of Tuscany (Italy), that had been previously identified based on morphological traits. The DNAs of two *N. campestris* from our sample were used as a negative control. Thermal cycling consisted of 3 min at 95 °C for initial denaturation, 35 cycles of denaturation at 95 °C for 30 s, annealing at 60 °C for 30 s, extension at 72 °C for 1 min, followed by 30 min at 72 °C for the final extension. The obtained PCR products were purified using FastAP and Exonuclease I (Thermo Scientific, Waltham, MA, USA) according to the manufacturer’s protocol and subjected to direct Sanger sequencing in forward and reverse or purified from 1% agarose (Invitrogen, Waltham, MA, USA) gels using the ReliaPrep DNA Clean-Up and Concentration System (Promega, Madison, WI, USA ), and ligated, propagated and PCR amplified as described by Marchi et al. [20]. At least five inserts per insect were sequenced in forward and reverse using T7/SP6 plasmid primers and chromatograms visualized using SnapGene (Dotmatics, Boston, MA, USA). Identity searches were performed on the GenBank database.

### 4.3. DNA Extraction

DNA extraction from the target (adult specimens) and non-target samples (adults and preimaginal stages) was performed according to the protocol already described by Rizzo et al. [21]. Each insect specimen was ground and homogenized individually using nylon mesh U-shaped bags (Bioreba, Reinach, Switzerland). Then we added 1 mL of 2% CTAB (cetyltrimethylammonium bromide) buffer (2% CTAB, 1% PVP-40, 100 mM Tris–HCl, pH 8.0, 1.4MNaCl, 20 mM EDTA, and 1% sodium metabisulfite). Then, 0.5 mL of lysate was incubated at 65 °C for 10 min. One volume of chloroform was added, stirred by inversion, and centrifuged at 13,000 rpm for 5 min. After this step, 600 μL of the upper phase were purified using the Maxwell RSC PureFood GMO kit and the Authentication Kit (Promega, Madison, WI, USA) according to the manufacturer’s protocol (catalog number selected: AS1600).

Each sample was extracted in duplicate, and DNA was eluted in 100 µL of nuclease-free water and either used in qPCR immediately or stored at −20 °C until use. The quality and quantity of extracted DNA were assayed using a QiaExpert spectrophotometer (Qiagen, Hilden, DE, Germany). The qualitative assessment detected the presence of inhibitors measuring the optical density ratio of A260/230 and A260/280 for diluted and undiluted DNA extracts. Moreover, to evaluate qualitatively the extracted DNA and its suitability for qPCR assays, the DNA samples were first amplified using the 18S uni-F/18S uni-R primers (5′-GCAAGGCTGAAACTTAAAGGAA-3′/5′-CCACCACCCATAGAATCAAGA-3′) and the 18S uni-P probe (HEX-ACGGAAGGGCACCACCAGGAGT-BHQ1) targeting the highly conserved region on the 18S ribosomal DNA [9].

### 4.4. Design of P. italosignus Primers

*P. italosignus*-specific primer pair, Pital_348F/Pital_465R, targeting a 119 bp region of the EF-1αgene sequence of *P. italosignus* Pi2-1 was designed using the *OligoArchitect^TM^ Primers and Probe Design Online software* (Sigma-Aldrich, St. Louis, MO, USA), taking into account primer melting temperatures, amplicon length, and the absence of secondary structures (Table 6). The oligos were synthesized by Eurofins Genomics (Ebersberg, Germany). The primer specificity was analyzed by the software BLAST (https://blast.ncbi.nlm.nih.gov/Blast.cgi; accessed on 1 December 2021), searching homologous nucleotide sequences and aligning them by the MAFFT software package [22] implemented within the software Geneious® 10.2.4 (Biomatters, http://www.geneious.com; accessed on 1 December 2021).

### 4.5. QPCR Optimization

The optimal conditions for qPCR were determined early by defining the optimal annealing temperature and the best oligo concentration. The primer annealing amplification temperature was found to be between 52 and 60 °C. The oligos were tested at three different concentrations (0.2, 0.3, and 0.4 µM) to determine the optimal one. The qPCR amplification reactions were performed in a CFX96 thermocycler (Biorad, Hercules, CA, USA) at a final volume of the reaction mixture of 20 μL. The reactions took place in a 96-well plate for real-time PCR (Starlab, Milano, Italy) in a vessel with a volume of 0.2 mL; each reaction was performed twice. For each run, two tubes containing 2 μL dd-water were tested as No Template Controls (NTCs). A positive and a negative amplification control were included for the target samples: *P. italosignus* (PAC) and *P. spumarius* (NAC), respectively. Tests were repeated in case of unclear or contradictory results. The data were analyzed using CFX Maestro 1.0 (Biorad, Hercules, CA, USA).

### 4.6. Performance Characteristics

According to PM7/98 (5) [23], the assay validation was performed on the analytical specificity, sensitivity, repeatability, and reproducibility. The analytical specificity was tested by comparing the qPCR amplification on the target and non-target samples, using extracted genomic DNA, at a final concentration of 10 ng/µL, from target and non-target organisms as well as source samples.

The analytical sensitivity (the limit of detection, or LoD) was evaluated using a 10-fold 1:5 serial dilution in triplicate from genomic DNA extracts (10 ng/µL) from single adults. The evaluation range was from 10 ng/µL to 2.56 × 10^−5^ ng/µL. All measurements were made using the QIAxpert (Qiagen, Hilden, Germany). Mean Cq values and standard deviations (SD) were calculated for the target species.

The repeatability of the assay was tested on 10 DNA samples of *P. italosignus* adults, diluted at a concentration of 5 ng/µL. The reproducibility was tested by a different operator performing the protocol on a different day. The raw data of qPCR amplification were analyzed using the CFX Maestro 2.3 software.

### 4.7. Insect Sampling for Protocol Validation

To validate our protocol, further insect samplings were carried out in the Municipality of Orbetello (province of Grosseto), where a population of *P. italosignus* had been recently found, with the same aforementioned methodology. This new area (42.416836N; 11.298914E) is about 30 km south of the Maremma Regional Park, where the original *P. italosignus* population had been first recorded, and it is characterized by a vegetation assemblage, which includes the lily *A. ramosus*, *Cistus monspeliensis* L., olive trees, Mediterranean cypresses, holm oaks, and other plants of the Mediterranean Maquis. Samplings were carried out on December 2021 and June 2022. After morphological analysis, DNA from all *Philaenus* adult specimens was used to validate our molecular diagnostic protocol.

## Figures and Tables

**Figure 1 plants-11-03314-f001:**
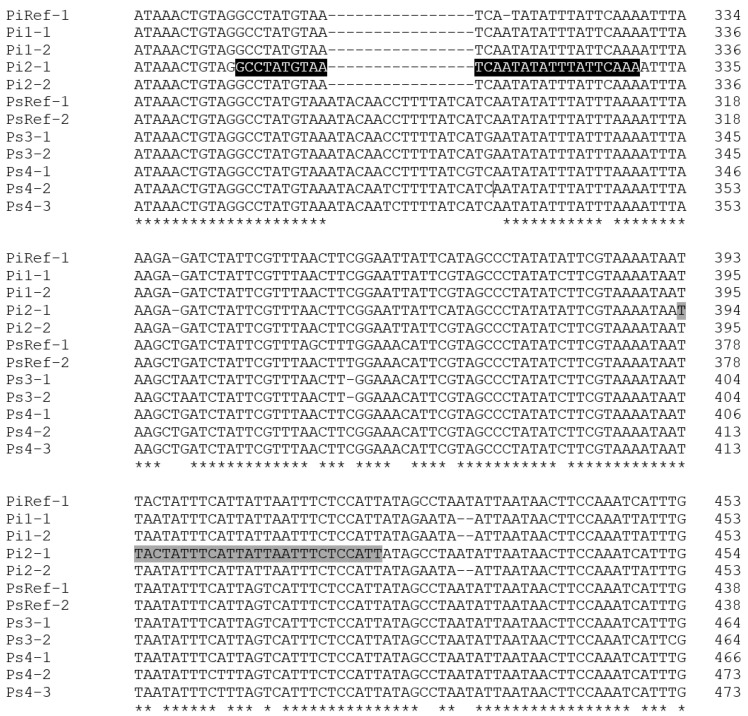
Partial alignment of the intronic regions of nine EF1-α alleles cloned from two *P. italosignus* (Pi1 and Pi2) and two *P. spumarius* (Ps3 and Ps4) specimens collected during this study. The homologous sequences of the two alleles of *P. spumarius* PSPU_08 (GCA_018207615.1) and the only allele of *P. italosignus* (JF309079) currently available in the Genbank database were included for comparative purposes as PsRef1, PsRef2, and PiRef1, respectively. The positions of the *P. italosignus*-specific qPCR assay primers, Pital_348F and Pital_465R, are highlighted in black and gray, respectively (the complete alignment of Spit-Uni1/2 products is shown in Appendix A). The “*” character indicates positions that are fully conserved.

**Figure 2 plants-11-03314-f002:**
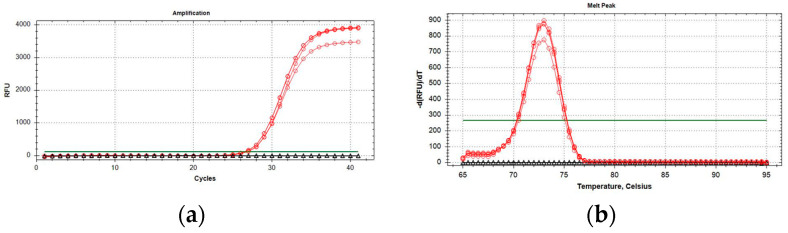
Amplification curves (**a**) and melting peaks (**b**) of *P. italosignus* adults (red circles). Black triangles: adults of *N. campestris* and *P. spumarius*.

**Figure 3 plants-11-03314-f003:**
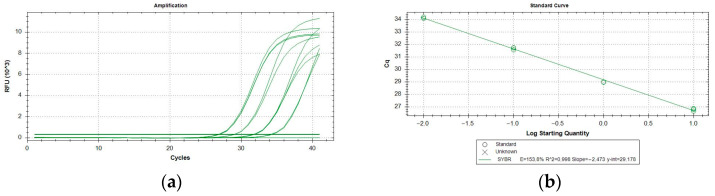
*P. italosignus*: serial dilutions 1:5 from 10 ng of adult DNA extracts, showing amplification (**a**) and titration curves (**b**).

**Table 1 plants-11-03314-t001:** Average concentrations of the extracted DNA (± SD), absorbance ratio (A260/280), and Cq values for assayed *Philaenus italosignus* adults.

Parameter	Adults (n * = 12)
DNA conc (ng/µL) ± SD	78.00 ± 3.20
(A260/280)	1.85 ± 0.24
Cq (18S)	19.75 ± 1.80

* sample size.

**Table 2 plants-11-03314-t002:** *P. italosignus* analytical sensitivity (LoD) assays using 1:5 serial dilutions (from 10 ng/µL to 25.6 fg/µL). The Cq values are the mean of the three threshold cycles of each dilution; Cq values above 35 were considered negative results.

Dilutions 1:5 (ng/µL)	Adult
	Cq Means *±* SD
10	27.15 ± 0.07
2.0	29.47 ± 0.29
0.4	32.09 ± 0.06
0.08	34.81 ± 0.55
0.016	35.40 ± 0.14
3.2 × 10^−3^	-
6.4 × 10^−4^	-
1.28 × 10^−4^	-
2.56 × 10^−5^	-

**Table 3 plants-11-03314-t003:** *P. italosignus*: repeatability and reproducibility values (average ± SD) of the assay.

N°	Repeatability		Reproducibility	
	Assay 1 (Average ± SD)	Assay 2 (Average ± SD)	Assay 1 (Average ± SD)	Assay 2 (Average ± SD)
1	26.88 ± 0.15	26.94 ± 0.24	26.58 ± 0.21	26.98 ± 0.28
2	26.63 ± 0.07	26.57 ± 0.01	26.88 ± 0.05	26.80 ± 0.06
3	26.60 ± 0.14	26.70 ± 0.13	26.49 ± 0.24	26.52 ± 0.24
4	26.74 ± 0.16	26.81 ± 0.16	26.45 ± 0.06	26.48 ± 0.06
5	26.71 ± 0.19	26.87 ± 0.07	26.45 ± 0.20	26.43 ± 0.20
6	26.62 ± 0.15	26.69 ± 0.14	26.50 ± 0.11	26.49 ± 0.11
7	26.68 ± 0.14	26.75 ± 0.21	26.52 ± 0.02	26.40 ± 0.02
8	26.71 ± 0.09	26.76 ± 0.03	26.47 ± 0.04	26.63 ± 0.04
9	26.66 ± 0.09	26.72 ± 0.06	26.63 ± 0.10	26.43 ± 0.10
10	26.74 ± 0.11	26.77 ± 0.18	26.68 ± 0.03	26.42 ± 0.03

**Table 4 plants-11-03314-t004:** Summary of field experiment samplings regarding *Philaenus* spp. For every specimen, the results of our qPCR assay were never conflicting with our preliminary morphological analysis.

Sampling Period	*P. spumarius*			*P. italosignus*		
	Females	Males	Total	Females	Males	Total
December 2021	10	5	15	2	0	2
June 2022	0	1	1	14	11	25

**Table 5 plants-11-03314-t005:** List of samples (target specimens from the Maremma Regional Park and non-target specimens from other Tuscan sites) used to set up the molecular diagnostic assay. RPS: Regional Phytopathological Service.

Order	Family	Species	Origin of Samples	Number of Specimens Assayed	Life Stage
Hemiptera	Pentatomidae	*Rhaphigaster nebulosa* (Poda, 1761)	RPS—Florence	1	Adult
	Tingidae	*Stephanitis lauri* Rietschel, 2014	University of Pisa	1	Adult
	Ricaniidae	*Ricania speculum* (Walker, 1851)	University of Pisa	1	Adult
	Cicadellidae	*Cicadella viridis* (Linnaeus, 1758)	RPS—Florence	1	Adult
		*Synophropsis lauri* (Horvath, 1897)	University of Florence	1	Adult
	Membracidae	*Stictocephala bisonia* Kopp &Yonke, 1977	University of Florence	1	Adult
	Aphrophoridae	*Philaenus spumarius* (Linnaeus, 1758)	University of Florence	37	Adult
		*Philaenus italosignus* Drosopoulos & Remane, 2000	University of Florence	12	Adult
		*Neophilaenus campestris* (Fallen, 1805)	University of Florence	75	Adult
	Cercopidae	*Lepyronia coleoptrata* (Linnaeus, 1758)	University of Florence	3	Adult
		*Cercopis sanguinolenta* (Scopoli, 1763)	University of Florence	3	Adult
		*Cercopis vulnerata* Rossi, 1807	University of Florence	3	Adult
	Dictyopharidae	*Dictyophara europaea* (Linnaeus, 1767)	RPS—Florence	1	Adult
	Aleyrodidae	*Dialeurodes citri* (Ashmead, 1885)	RPS—Florence	1	Juvenile
Lepidoptera	Crambidae	*Cydalima perspectalis* (Walker, 1859)	RPS—Florence	1	Larva
	Tortricidae	*Grapholita (Aspila) molesta* (Busck, 1916)	University of Florence	1	Adult
		*Cydia pomonella* (Linnaeus, 1758)	University of Florence	1	Adult
		*Cryptoblabes gnidiella* (Millière, 1867)	University of Florence	1	Adult
Diptera	Tephritidae	*Ceratitis capitata* (Wiedemann 1824)	University of Florence	2	Adult
		*Ceratitis capitata* (Wiedemann 1824)	University of Florence	2	Larva
		*Rhagoletis cerasi* (Linnaeus, 1758)	University of Florence	1	Pupa
		*Rhagoletis completa* Cresson, 1929	RPS—Florence	1	Larva
		*Acanthiophilus helianthi* (Rossi, 1794)	University of Pisa	1	Adult

**Table 6 plants-11-03314-t006:** List of the primers used in the SYBR Green assay.

Primer Name	Length (Nucleotides)	Sequence 5′–3′	Product Size (bp)
Pital_348F	28	GCCTATGTAATCAATATATTTATTCAAA	119
Pital_465R	28	AATGGAGAAATTAATAATGAAATAGTAA	

## Data Availability

Most of the data presented in this study are contained within the article and in Appendix A. Data not shown in the article are available on request from the corresponding author.

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
