# Peer review of "Quantitative Real-Time PCR Based on SYBR Green Technology for the Identification of Philaenus italosignus Drosopoulos & Remane (Hemiptera Aphrophoridae)"

_plants, 2022, doi:10.3390/plants11233314_

Round 1

Author Response

Line numbers in our responses refer to the revised manuscript.

Reviewer 1

There is a lack of reference to the cost to an analytical laboratory of adopting the suggested method.

Done, see lines 163-166.

Suggestions: State (e.g., at the end of “Discussion”) the perspective to expand the number of samples on which to test and validate the protocol developed.

Done, see lines 196-198

General Notes:

  • In the list of author’s names, superscript 3 is before the dot.

done

  • In the abstract, Philaenus spumarius is cited directly with a dotted "P," never in full. Even if the genus has already been cited with another species, the first time it is mentioned, it should be put in full.

done

Specific Comments:

R73 Specify what the "Cq value" is.

done

R88-89 Specify that insertions/deletions and indels are synonyms.

done

R148 Indicate, even if only in the supplementary material, the results of the preliminary morphological analysis.

The results of our preliminary morphological analysis are already stated at lines 143-149.

R159-160 The molecular assay, in my opinion, is also useful because there are not many insects to test, and its use means that samples are not wasted.

Agreed. In fact a low number of samples may result in an even lower number of male adults. In that case molecular tests may exploit all specimens, even females that cannot be as easily determined morphologically.
A few sentence were added accordingly. See lines 168-172.

R181 "in" is missing before "different seasons."

done

R287 "two tubes containing two μL": I would write the second "two" as a number to conform to the style used in the previous lines and differentiate it from the one referring to tubes.

done

R289 Clarify which are the positive and negative amplification control included in the target samples.

done

R306 "Different day" is more elegant than "Extra day."

done

Reviewer 2 Report

This is a carefully done study and the findings are of considerable interest. A few minor revisions are listed below.

Table 1: The number of samples should be clearly stated.

Table 3 is not cited. It is probably at the end of the 125th line of the sentence.

Figure 1 and Table 6 should be related. The sequence of Pital_465R is different in Figure 1 and Table 6. Four bases on the 3' side of the Pital_465R primer differ between Figure 1 and Table 6.

Line 114: The title should be prefixed with "2.3".

Line 135: The title should be prefixed with "2.4".

Although it is not necessary to conduct a follow-up study, it may be worth mentioning that the real-time PCR of this study is capable of discriminating individuals in life stages other than adult.

Author Response

Line numbers in our responses refer to the revised manuscript.

Reviewer 2

Table 1: The number of samples should be clearly stated.

done

Table 3 is not cited. It is probably at the end of the 125th line of the sentence.

Done, also we noticed a missing reference to table 6 so we added that at line 285.

Figure 1 and Table 6 should be related. The sequence of Pital_465R is different in Figure 1 and Table 6. Four bases on the 3' side of the Pital_465R primer differ between Figure 1 and Table 6.

There was a typo involving the sample used for the designing of the primers that led to the wrong highlighting of the alignment in Figure 1. In fact, the sample chosen to design the primers was Pi2-1. We changed the text and the figure accordingly.No mismatch is now present between table 6 and Figure 1. However the resulting mismatch involved only position 398; note that for the primer reverse, in figure 1, the first base in 3’ side is on the previous section of the alignment. Also the supplementary material has been updated accordingly.

Line 114: The title should be prefixed with "2.3".

done

Line 135: The title should be prefixed with "2.4".

done

Although it is not necessary to conduct a follow-up study, it may be worth mentioning that the real-time PCR of this study is capable of discriminating individuals in life stages other than adult.

Done, see lines 196-198.
